# How Much Restricted Isometry is Needed In Nonconvex Matrix Recovery?

**Richard Y. Zhang**
University of California, Berkeley
ryz@alum.mit.edu

**Cédric Josz**
University of California, Berkeley
cedric.josz@gmail.com

**Somayeh Sojoudi**
University of California, Berkeley
sojoudi@berkeley.edu

**Javad Lavaei**
University of California, Berkeley
lavaei@berkeley.edu

## Abstract

When the linear measurements of an instance of low-rank matrix recovery satisfy a restricted isometry property (RIP)—i.e. they are approximately norm-preserving—the problem is known to contain *no spurious local minima*, so exact recovery is guaranteed. In this paper, we show that moderate RIP is not enough to eliminate spurious local minima, so existing results can only hold for near-perfect RIP. In fact, counterexamples are ubiquitous: we prove that every $x$ is the spurious local minimum of a rank-1 instance of matrix recovery that satisfies RIP. One specific counterexample has RIP constant $\delta = 1/2$, but causes randomly initialized stochastic gradient descent (SGD) to fail 12% of the time. SGD is frequently able to avoid and escape spurious local minima, but this empirical result shows that it can occasionally be defeated by their existence. Hence, while exact recovery guarantees will likely require a proof of *no spurious local minima*, arguments based solely on norm preservation will only be applicable to a narrow set of nearly-isotropic instances.

## 1 Introduction

Recently, several important nonconvex problems in machine learning have been shown to contain *no spurious local minima* [19, 4, 21, 8, 20, 34, 30]. These problems are easily solved using local search algorithms despite their nonconvexity, because every local minimum is also a global minimum, and every saddle-point has sufficiently negative curvature to allow escape. Formally, the usual first- and second-order necessary conditions for local optimality (i.e. zero gradient and a positive semidefinite Hessian) are also *sufficient* for global optimality; satisfying them to $\epsilon$-accuracy will yield a point within an $\epsilon$-neighborhood of a globally optimal solution.

Many of the best-understood nonconvex problems with no spurious local minima are variants of the *low-rank matrix recovery* problem. The simplest version (known as *matrix sensing*) seeks to recover an $n \times n$ positive semidefinite matrix $Z$ of low rank $r \ll n$, given measurement matrices $A_1, \ldots, A_m$ and noiseless data $b_i = \langle A_i, Z \rangle$. The usual, nonconvex approach is to solve the following

$$\underset{x \in \mathbb{R}^{n \times r}}{\text{minimize}} \ \|\mathcal{A}(xx^T) - b\|^2 \quad \text{where} \quad \mathcal{A}(X) = [\langle A_1, X \rangle \quad \cdots \quad \langle A_m, X \rangle]^T \qquad (1)$$

to second-order optimality, using a local search algorithm like (stochastic) gradient descent [19, 24] and trust region Newton's method [16, 7], starting from a random initial point.

Exact recovery of the ground truth $Z$ is guaranteed under the assumption that $\mathcal{A}$ satisfies the *restricted isometry property* [14, 13, 31, 11] with a sufficiently small constant. The original result is

due to Bhojanapalli et al. [4], though we adapt the statement below from a later result by Ge et al. [20, Theorem 8]. (Zhu et al. [43] give an equivalent statement for nonsymmetric matrices.)

**Definition 1** (Restricted Isometry Property). The linear map $\mathcal{A} : \mathbb{R}^{n \times n} \to \mathbb{R}^m$ is said to satisfy $(r, \delta_r)$-RIP with constant $0 \leq \delta_r < 1$ if there exists a fixed scaling $\gamma > 0$ such that for all rank-$r$ matrices $X$:

$$(1 - \delta_r)\|X\|_F^2 \leq \gamma \cdot \|\mathcal{A}(X)\|^2 \leq (1 + \delta_r)\|X\|_F^2. \tag{2}$$

We say that $\mathcal{A}$ satisfies $r$-RIP if $\mathcal{A}$ satisfies $(r, \delta_r)$-RIP with some $\delta_r < 1$.

**Theorem 2** (No spurious local minima). *Let $\mathcal{A}$ satisfy $(2r, \delta_{2r})$-RIP with $\delta_{2r} < 1/5$. Then, (1) has no spurious local minima: every local minimum $x$ satisfies $xx^T = Z$, and every saddle point has an escape (the Hessian has a negative eigenvalue). Hence, any algorithm that converges to a second-order critical point is guaranteed to recover $Z$ exactly.*

Standard proofs of Theorem 2 use a *norm-preserving* argument: if $\mathcal{A}$ satisfies $(2r, \delta_{2r})$-RIP with a small constant $\delta_{2r}$, then we can view the least-squares residual $\mathcal{A}(xx^T) - b$ as a dimension-reduced embedding of the displacement vector $xx^T - Z$, as in

$$\|\mathcal{A}(xx^T) - b\|^2 = \|\mathcal{A}(xx^T - Z)\|^2 \approx \|xx^T - Z\|_F^2 \text{ up to scaling.} \tag{3}$$

The high-dimensional problem of minimizing $\|xx^T - Z\|_F^2$ over $x$ contains no spurious local minima, so its dimension-reduced embedding (1) should satisfy a similar statement. Indeed, this same argument can be repeated for noisy measurements and nonsymmetric matrices to result in similar guarantees [4, 20].

The norm-preserving argument also extends to "harder" choices of $\mathcal{A}$ that do not satisfy RIP over its entire domain. In the matrix completion problem, the RIP-like condition $\|\mathcal{A}(X)\|^2 \approx \|X\|_F^2$ holds only when $X$ is both low-rank and sufficiently dense [12]. Nevertheless, Ge et al. [21] proved a similar result to Theorem 2 for this problem, by adding a regularizing term to the objective. For a detailed introduction to the norm-preserving argument and its extension with regularizers, we refer the interested reader to [21, 20].

## 1.1 How much restricted isometry?

The RIP threshold $\delta_{2r} < 1/5$ in Theorem 2 is highly conservative—it is only applicable to nearly-isotropic measurements like Gaussian measurements. Let us put this point into perspective by measuring distortion using the *condition number*[1] $\kappa_{2r} \in [1, \infty)$. Deterministic linear maps from real-life applications usually have condition numbers $\kappa_{2r}$ between $10^2$ and $10^4$, and these translate to RIP constants $\delta_{2r} = (\kappa_{2r} - 1)/(\kappa_{2r} + 1)$ between 0.99 and 0.9999. By contrast, the RIP threshold $\delta_{2r} < 1/5$ requires an equivalent condition number of $\kappa_{2r} = (1 + \delta_{2r})/(1 - \delta_{2r}) < 3/2$, which would be considered *near-perfect* in linear algebra.

In practice, nonconvex matrix completion works for a much wider class of problems than those suggested by Theorem 2 [6, 5, 32, 1]. Indeed, assuming only that $\mathcal{A}$ satisfies $2r$-RIP, solving (1) to global optimality is enough to guarantee exact recovery [31, Theorem 3.2]. In turn, stochastic algorithms like stochastic gradient descent (SGD) are often able to attain global optimality. This disconnect between theory and practice motivates the following question.

**Can Theorem 2 be substantially improved—is it possible to guarantee the inexistence of spurious local minima with $(2r, \delta_{2r})$-RIP and any value of $\delta_{2r} < 1$?**

At a basic level, the question gauges the generality and usefulness of RIP as a base assumption for nonconvex recovery. Every family of measure operators $\mathcal{A}$—even correlated and "bad" measurement ensembles—will eventually come to satisfy $2r$-RIP as the number of measurements $m$ grows large. Indeed, given $m \geq n(n + 1)/2$ linearly independent measurements, the operator $\mathcal{A}$ becomes invertible, and hence trivially $2r$-RIP. In this limit, recovering the ground truth $Z$ from noiseless measurements is as easy as solving a system of linear equations. Yet, it remains unclear whether nonconvex recovery is guaranteed to succeed.

At a higher level, the question also gauges the wisdom of exact recovery guarantees through "no spurious local minima". It may be sufficient but not necessary; exact recovery may actually hinge

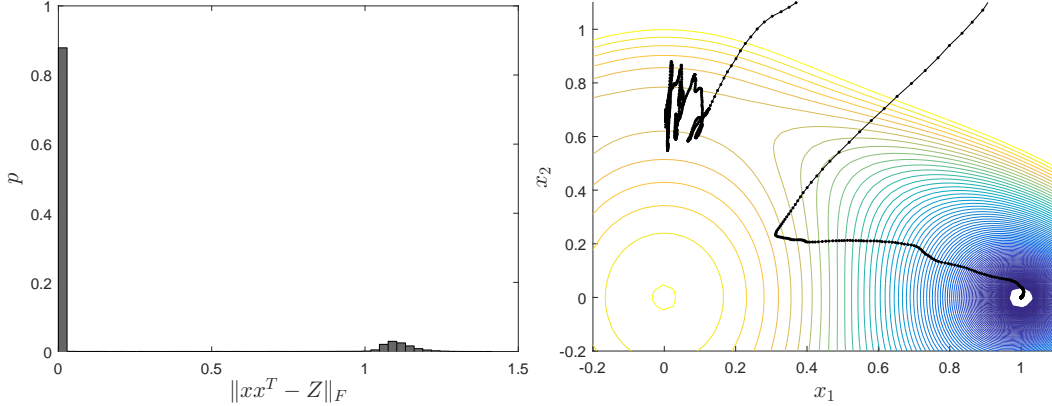

Figure 1: Solving Example 3 using stochastic gradient descent randomly initialized with the standard Gaussian. **(Left)** Histogram over 100,000 trials of final error $\|xx^T - Z\|_F$ after $10^3$ steps with learning rate $\alpha = 10^{-3}$ and momentum $\beta = 0.9$. **(Right)** Two typical stochastic gradient descent trajectories, showing convergence to the spurious local minimum at $(0, 1/\sqrt{2})$, and to the ground truth at $(1, 0)$.

on SGD's ability to avoid and escape spurious local minima when they do exist. Indeed, there is growing empirical evidence that SGD outmaneuvers the "optimization landscape" of nonconvex functions [6, 5, 27, 32, 1], and achieves some global properties [22, 40, 39]. It remains unclear whether the success of SGD for matrix recovery should be attributed to the inexistence of spurious local minima, or to some global property of SGD.

## 1.2 Our results

In this paper, we give a strong negative answer to the question above. Consider the counterexample below, which satisfies $(2r, \delta_{2r})$-RIP with $\delta_{2r} = 1/2$, but nevertheless contains a spurious local minimum that causes SGD to fail in 12% of trials.

**Example 3.** Consider the following $(2, 1/2)$-RIP instance of (1) with matrices

$$Z = \begin{bmatrix} 1 & 0 \\ 0 & 0 \end{bmatrix}, \quad A_1 = \begin{bmatrix} \sqrt{2} & 0 \\ 0 & 1/\sqrt{2} \end{bmatrix}, \quad A_2 = \begin{bmatrix} 0 & \sqrt{3/2} \\ \sqrt{3/2} & 0 \end{bmatrix}, \quad A_3 = \begin{bmatrix} 0 & 0 \\ 0 & \sqrt{3/2} \end{bmatrix}.$$

Note that the associated operator $\mathcal{A}$ is invertible and satisfies $\|X\|_F^2 \leq \|\mathcal{A}(X)\|^2 \leq 3\|X\|_F^2$ for all $X$. Nevertheless, the point $x = (0, 1/\sqrt{2})$ satisfies second-order optimality,

$$f(x) \equiv \|\mathcal{A}(xx^T - Z)\|^2 = \frac{3}{2}, \qquad \nabla f(x) = \begin{bmatrix} 0 \\ 0 \end{bmatrix}, \qquad \nabla^2 f(x) = \begin{bmatrix} 0 & 0 \\ 0 & 8 \end{bmatrix},$$

and randomly initialized SGD can indeed become stranded around this point, as shown in Figure 1. Repeating these trials 100,000 times yields 87,947 successful trials, for a failure rate of $12.1 \pm 0.3\%$ to three standard deviations.

Accordingly, RIP-based exact recovery guarantees like Theorem 2 cannot be improved beyond $\delta_{2r} < 1/2$. Otherwise, spurious local minima can exist, and SGD may become trapped. Using a local search algorithm with a random initialization, "no spurious local minima" is not only sufficient for exact recovery, but also necessary.

In fact, there exists an infinite number of counterexamples like Example 3. In Section 3, we prove that, in the rank-1 case, *almost every* choice of $x, Z$ generates an instance of (1) with a strict spurious local minimum.

**Theorem 4** (Informal). *Let $x, z \in \mathbb{R}^n$ be nonzero and not colinear. Then, there exists an instance of (1) satisfying $(n, \delta_n)$-RIP with $\delta_n < 1$ that has $Z = zz^T$ as the ground truth and $x$ as a strict spurious local minimum, i.e. with zero gradient and a positive definite Hessian. Moreover, $\delta_n$ is*

*bounded in terms of the length ratio $\rho = \|x\|/\|z\|$ and the incidence angle $\phi$ satisfying $x^T z = \|x\|\|z\|\cos\phi$ as*

$$\delta_n \le \frac{\tau + \sqrt{1-\zeta^2}}{\tau + 1} \qquad where\ \zeta = \frac{\sin^2\phi}{\sqrt{(\rho^2-1)^2 + 2\rho^2\sin^2\phi}}, \quad \tau = \frac{2\sqrt{\rho^2 + \rho^{-2}}}{\zeta^2}$$

It is therefore impossible to establish "no spurious local minima" guarantees unless the RIP constant $\delta$ is small. This is a strong negative result on the generality and usefulness of RIP as a base assumption, and also on the wider norm-preserving argument described earlier in the introduction. In Section 4, we provide strong empirical evidence for the following *sharp* version of Theorem 2.

**Conjecture 5.** *Let $\mathcal{A}$ satisfy $(2r, \delta_{2r})$-RIP with $\delta_{2r} < 1/2$. Then, (1) has no spurious local minima. Moreover, the figure of $1/2$ is sharp due to the existence of Example 3.*

How is the *practical* performance of SGD affected by spurious local minima? In Section 5, we apply randomly initialized SGD to instances of (1) engineered to contain spurious local minima. In one case, SGD recovers the ground truth with a 100% success rate, as if the spurious local minima did not exist. But in another case, SGD fails in 59 of 1,000 trials, for a positive failure rate of $5.90 \pm 2.24\%$ to three standard deviations. Examining the failure cases, we observe that SGD indeed becomes trapped around a spurious local minimum, similar to Figure 1 in Example 3.

## 1.3  Related work

There have been considerable recent interest in understanding the empirical "hardness" of nonconvex optimization, in view of its well-established theoretical difficulties. Nonconvex functions contain saddle points and spurious local minima, and local search algorithms may become trapped in them. Recent work have generally found the matrix sensing problem to be "easy", particularly under an RIP-like incoherence assumption. Our results in this paper counters this intuition, showing—perhaps surprisingly—that the problem is generically "hard" even under RIP.

**Comparison to convex recovery.** Classical theory for the low-rank matrix recovery problem is based on convex relaxation: replacing $xx^T$ in (1) by a convex term $X \succeq 0$, and augmenting the objective with a trace penalty $\lambda \cdot \operatorname{tr}(X)$ to induce a low-rank solution [12, 31, 15, 11]. The convex approach enjoys RIP-based exact recovery guarantees [11], but these are also fundamentally restricted to small RIP constants [10, 38]—in direct analogy with our results for nonconvex recovery. In practice, convex recovery is usually much more expensive than nonconvex recovery, because it requires optimizing over an $n \times n$ matrix variable instead of an $n \times r$ vector-like variable. On the other hand, it is statistically consistent [3], and guaranteed to succeed with $m \ge \frac{1}{2}n(n+1)$ noiseless, linearly independent measurements. By comparison, our results show that nonconvex recovery can still fail in this regime.

**Convergence to spurious local minima.** Recent results on "no spurious local minima" are often established using a norm-preserving argument: the problem at hand is the low-dimension embedding of a canonical problem known to contain no spurious local minima [19, 34, 35, 4, 21, 20, 30, 43]. While the approach is widely applicable in its scope, our results in this paper finds it to be restrictive in the problem data. More specifically, the measurement matrices $A_1, \ldots, A_m$ must come from a nearly-isotropic ensemble like the Gaussian and the sparse binary.

**Special initialization schemes.** An alternative way to guarantee exact recovery is to place the initial point sufficiently close to the global optimum [25, 26, 23, 42, 41, 36]. This approach is more general because it does not require a global "no spurious local minima" guarantee. On the other hand, good initializations are highly problem-specific and difficult to generalize. Our results show that spurious local minima can exist arbitrarily close to the solution. Hence, exact recovery guarantees must give proof of local attraction, beyond simply starting close to the ground truth.

**Ability of SGD to escape spurious local minima.** Practitioners have long known that stochastic gradient descent (SGD) enjoys properties inherently suitable for the sort of nonconvex optimization problems that appear in machine learning [27, 6], and that it is well-suited for generalizing unseen data [22, 40, 39]. Its specific behavior is yet not well understood, but it is commonly conjectured that SGD outperforms classically "better" algorithms like BFGS because it is able to avoid and escape spurious local minima. Our empirical findings in Section 5 partially confirms this suspicion,

showing that randomly initialized SGD is sometimes able to avoid and escape spurious local minima as if they did not exist. In other cases, however, SGD can indeed become stuck at a local minimum, thereby resulting in a positive failure rate.

**Notation**

We use $x$ to refer to any candidate point, and $Z = zz^T$ to refer to a rank-$r$ factorization of the ground truth $Z$. For clarity, we use lower-case $x, z$ even when these are $n \times r$ matrices.

The sets $\mathbb{R}^{n \times n} \supset \mathbb{S}^n$ are the space of $n \times n$ real matrices and real symmetric matrices, and $\langle X, Y \rangle \equiv \mathrm{tr}(X^T Y)$ and $\|X\|_F^2 \equiv \langle X, X \rangle$ are the Frobenius inner product and norm. We write $X \succeq 0$ (resp. $X \succ 0$) if $X$ is positive semidefinite (resp. positive definite). Given a matrix $M$, its spectral norm is $\|M\|$, and its eigenvalues are $\lambda_1(M), \ldots, \lambda_n(M)$. If $M = M^T$, then $\lambda_1(M) \geq \cdots \geq \lambda_n(M)$ and $\lambda_{\max}(M) \equiv \lambda_1(M)$, $\lambda_{\min}(M) \equiv \lambda_n(M)$. If $M$ is invertible, then its condition number is $\mathrm{cond}(M) = \|M\|\|M^{-1}\|$; if not, then $\mathrm{cond}(M) = \infty$.

The vectorization operator $\mathrm{vec} : \mathbb{R}^{n \times n} \to \mathbb{R}^{n^2}$ preserves inner products $\langle X, Y \rangle = \mathrm{vec}(X)^T \mathrm{vec}(Y)$ and Euclidean norms $\|X\|_F = \|\mathrm{vec}(X)\|$. In each case, the matricization operator $\mathrm{mat}(\cdot)$ is the inverse of $\mathrm{vec}(\cdot)$.

## 2  Key idea: Spurious local minima via convex optimization

Given arbitrary $x \in \mathbb{R}^{n \times r}$ and rank-$r$ positive semidefinite matrix $Z \in \mathbb{S}^n$, consider the problem of finding an instance of (1) with $Z$ as the ground truth and $x$ as a spurious local minimum. While not entirely obvious, this problem is actually convex, because the first- and second-order optimality conditions associated with (1) are *linear matrix inequality* (LMI) constraints [9] with respect to the *kernel* operator $\mathcal{H} \equiv \mathcal{A}^T \mathcal{A}$. The problem of finding an instance of (1) that also satisfies RIP is indeed nonconvex. However, we can use the *condition number* of $\mathcal{H}$ as a surrogate for the RIP constant $\delta$ of $\mathcal{A}$: if the former is finite, then the latter is guaranteed to be less than 1. The resulting optimization is convex, and can be numerically solved using an interior-point method, like those implemented in SeDuMi [33], SDPT3 [37], and MOSEK [2], to high accuracy.

We begin by fixing some definitions. Given a choice of $\mathcal{A} : \mathbb{S}^n \to \mathbb{R}^m$ and the ground truth $Z = zz^T$, we define the nonconvex objective

$$f : \mathbb{R}^{n \times r} \to \mathbb{R} \qquad \text{such that} \qquad f(x) = \|\mathcal{A}(xx^T - zz^T)\|^2 \qquad (4)$$

whose value is always nonnegative by construction. If the point $x$ attains $f(x) = 0$, then we call it a *global minimum*; otherwise, we call it a *spurious* point. Under RIP, $x$ is a global minimum if and only if $xx^T = zz^T$ [31, Theorem 3.2]. The point $x$ is said to be a *local minimum* if $f(x) \leq f(x')$ holds for all $x'$ within a local neighborhood of $x$. If $x$ is a local minimum, then it must satisfy the first and second-order *necessary* optimality conditions (with some fixed $\mu \geq 0$):

$$\langle \nabla f(x), u \rangle = 2 \langle \mathcal{A}(xx^T - zz^T), \mathcal{A}(xu^T + ux^T) \rangle = 0 \qquad \forall u \in \mathbb{R}^{n \times r}, \quad (5)$$

$$\langle \nabla^2 f(x)u, u \rangle = 2 \langle \mathcal{A}(xx^T - zz^T), uu^T \rangle + \|\mathcal{A}(xu^T + ux^T)\|^2 \geq \mu \|u\|_F^2 \quad \forall u \in \mathbb{R}^{n \times r}. \quad (6)$$

Conversely, if $x$ satisfies the second-order *sufficient* optimality conditions, that is (5)-(6) with $\mu > 0$, then it is a local minimum. Local search algorithms are only guaranteed to converge to a *first-order critical point* $x$ satisfying (5), or a *second-order critical point* $x$ satisfying (5)-(6) with $\mu \geq 0$. The latter class of algorithms include stochastic gradient descent [19], randomized and noisy gradient descent [19, 28, 24, 18], and various trust-region methods [17, 29, 16, 7].

Given arbitrary choices of $x, z \in \mathbb{R}^{n \times r}$, we formulate the problem of picking an $\mathcal{A}$ satisfying (5) and (6) as an LMI feasibility. First, we define $\mathbf{A} = [\mathrm{vec}(A_1), \ldots, \mathrm{vec}(A_m)]^T$ satisfying $\mathbf{A} \cdot \mathrm{vec}(X) = \mathcal{A}(X)$ for all $X$ as the matrix representation of the operator $\mathcal{A}$. Then, we rewrite (5) and (6) as $2 \cdot \mathscr{L}(\mathbf{A}^T \mathbf{A}) = 0$ and $2 \cdot \mathscr{M}(\mathbf{A}^T \mathbf{A}) \succeq \mu I$, where the linear operators $\mathscr{L}$ and $\mathscr{M}$ are defined

$$\mathscr{L} : \mathbb{S}^{n^2} \to \mathbb{R}^{n \times r} \qquad \text{such that} \qquad \mathscr{L}(\mathbf{H}) \equiv 2 \cdot \mathbf{X}^T \mathbf{H} e, \qquad (7)$$

$$\mathscr{M} : \mathbb{S}^{n^2} \to \mathbb{S}^{nr \times nr} \qquad \text{such that} \qquad \mathscr{M}(\mathbf{H}) \equiv 2 \cdot [I_r \otimes \mathrm{mat}(\mathbf{H}e)^T] + \mathbf{X}^T \mathbf{H} \mathbf{X}, \qquad (8)$$

with respect to the error vector $e = \mathrm{vec}(xx^T - zz^T)$ and the $n^2 \times nr$ matrix $\mathbf{X}$ that implements the symmetric product operator $\mathbf{X} \cdot \mathrm{vec}(u) = \mathrm{vec}(xu^T + ux^T)$. To compute a choice of $\mathbf{A}$ satisfying

$\mathscr{L}(\mathbf{A}^T\mathbf{A}) = 0$ and $\mathscr{M}(\mathbf{A}^T\mathbf{A}) \succeq 0$, we solve the following LMI feasibility problem

$$\underset{\mathbf{H}}{\text{maximize}} \quad 0 \qquad \text{subject to} \qquad \mathscr{L}(\mathbf{H}) = 0, \quad \mathscr{M}(\mathbf{H}) \succeq \mu I, \quad \mathbf{H} \succeq 0, \qquad (9)$$

and factor a feasible $\mathbf{H}$ back into $\mathbf{A}^T\mathbf{A}$, e.g. using Cholesky factorization or an eigendecomposition. Once a matrix representation $\mathbf{A}$ is found, we recover the matrices $A_1, \ldots, A_m$ implementing the operator $\mathcal{A}$ by matricizing each row of $\mathbf{A}$.

Now, the problem of picking $\mathcal{A}$ with the smallest condition number may be formulated as the following LMI optimization

$$\underset{\mathbf{H},\eta}{\text{maximize}} \quad \eta \qquad \text{subject to} \qquad \eta I \preceq \mathbf{H} \preceq I, \quad \mathscr{L}(\mathbf{H}) = 0, \quad \mathscr{M}(\mathbf{H}) \succeq \mu I, \quad \mathbf{H} \succeq 0, \quad (10)$$

with solution $\mathbf{H}^\star, \eta^\star$. Then, $1/\eta^\star$ is the best condition number achievable, and any $\mathcal{A}$ recovered from $\mathbf{H}^\star$ will satisfy

$$\left(1 - \frac{1-\eta^\star}{1+\eta^\star}\right)\|X\|^2 \leq \frac{2}{1+\eta^\star}\|\mathcal{A}(X)\|_F^2 \leq \left(1 + \frac{1-\eta^\star}{1+\eta^\star}\right)\|X\|^2$$

for all $X$, that is, with *any rank*. As such, $\mathcal{A}$ is $(n, \delta_n)$-RIP with $\delta_n = (1 - \eta^\star)/(1 + \eta^\star)$, and hence also $(p, \delta_p)$-RIP with $\delta_p \leq \delta_n$ for all $p \in \{1, \ldots, n\}$; see e.g. [31, 11]. If the optimal value $\eta^\star$ is strictly positive, then the recovered $\mathcal{A}$ yields an RIP instance of (1) with $zz^T$ as the ground truth and $x$ as a spurious local minimum, as desired.

It is worth emphasizing that a small condition number—a large $\eta^\star$ in (10)—will always yield a small RIP constant $\delta_n$, which then bounds all other RIP constants via $\delta_n \geq \delta_p$ for all $p \in \{1, \ldots, n\}$. However, the converse direction is far less useful, as the value of $\delta_n = 1$ does not preclude $\delta_p$ with $p < n$ from being small.

## 3 Closed-form solutions

It turns out that the LMI problem (10) in the rank-1 case is sufficiently simple that it can be solved in closed-form. (All proofs are given in the Appendix.) Let $x, z \in \mathbb{R}^n$ be arbitrary nonzero vectors, and define

$$\rho \equiv \frac{\|x\|}{\|z\|}, \qquad\qquad \phi \equiv \arccos\left(\frac{x^T z}{\|x\|\|z\|}\right), \qquad (11)$$

as their associated length ratio and incidence angle. We begin by examining the prevalence of spurious critical points.

**Theorem 6** (First-order optimality)**.** *The best-conditioned $\mathbf{H}^\star \succeq 0$ such that $\mathscr{L}(\mathbf{H}^\star) = 0$ satisfies*

$$\text{cond}(\mathbf{H}^\star) = \frac{1 + \sqrt{1 - \zeta^2}}{1 - \sqrt{1 - \zeta^2}} \qquad \text{where} \qquad \zeta \equiv \frac{\sin\phi}{\sqrt{(\rho^2 - 1)^2 + 2\rho^2 \sin^2\phi}}. \qquad (12)$$

*Hence, if $\phi \neq 0$, then $x$ is a first-order critical point for an instance of (1) satisfying $(2, \delta)$-RIP with $\delta = \sqrt{1 - \zeta^2} < 1$ given in (12).*

The point $x = 0$ is always a local maximum for $f$, and hence a spurious first-order critical point. With a perfect RIP constant $\delta = 0$, Theorem 6 says that $x = 0$ is also the only spurious first-order critical point. Otherwise, spurious first-order critical points may exist elsewhere, even when the RIP constant $\delta$ is arbitrarily close to zero. This result highlights the importance of converging to second-order optimality, in order to avoid getting stuck at a spurious first-order critical point.

Next, we examine the prevalence of spurious local minima.

**Theorem 7** (Second-order optimality)**.** *There exists $\mathbf{H}$ satisfying $\mathscr{L}(\mathbf{H}) = 0$, $\mathscr{M}(\mathbf{H}) \succeq \mu I$, and $\eta I \preceq \mathbf{H} \preceq I$ where*

$$\eta \geq \frac{1}{1+\tau} \cdot \left(\frac{1 + \sqrt{1 - \zeta^2}}{1 - \sqrt{1 - \zeta^2}}\right), \qquad \mu = \frac{\|z\|^2}{1+\tau}, \qquad \tau \equiv \frac{2\sqrt{\rho^2 + \rho^{-2}}}{\zeta^2}$$

*and $\zeta$ is defined in (12). Hence, if $\phi \neq 0$ and $\rho > 0$ is finite, then $x$ is a strict local minimum for an instance of (1) satisfying $(2, \delta)$-RIP with $\delta = (\tau + \sqrt{1 - \zeta^2})/(1 + \tau) < 1$.*

If $\phi \neq 0$ and $\rho > 0$, then $x$ is guaranteed to be a strict local minimum for a problem instance satisfying 2-RIP. Hence, we must conclude that spurious local minima are ubiquitous. The associated RIP constant $\delta < 1$ is not too much worse than than the figure quoted in Theorem 6. On the other hand, spurious local minima must cease to exist once $\delta < 1/5$ according to Theorem 2.

## 4 Experiment 1: Minimum $\delta$ with spurious local minima

What is smallest RIP constant $\delta_{2r}$ that still admits an instance of (1) with spurious local minima? Let us define the threshold value as the following

$$\delta^\star = \min_{x,Z,\mathcal{A}} \{\delta : \nabla f(x) = 0, \quad \nabla^2 f(x) \succeq 0, \quad \mathcal{A} \text{ satisfies } (2r, \delta)\text{-RIP}\}. \tag{13}$$

Here, we write $f(x) = \|\mathcal{A}(xx^T - Z)\|^2$, and optimize over the spurious local minimum $x \in \mathbb{R}^{n \times r}$, the rank-$r$ ground truth $Z \succeq 0$, and the linear operator $\mathcal{A} : \mathbb{R}^{n \times n} \to \mathbb{R}^m$. Note that $\delta^\star$ gives a "no spurious local minima" guarantee, due to the inexistence of counterexamples.

**Proposition 8.** *Let $\mathcal{A}$ satisfy $(2r, \delta_{2r})$-RIP. If $\delta_{2r} < \delta^\star$, then (1) has no spurious local minimum.*

*Proof.* Suppose that (1) contained a spurious local minimum $x$ for ground truth $Z$. Then, substituting this choice of $x, Z, \mathcal{A}$ into (13) would contradict the definition of $\delta^\star$ as the minimum. $\qquad\square$

Our convex formulation in Section 2 bounds $\delta^\star$ from above. Specifically, our LMI problem (10) with optimal value $\eta^\star$ is equivalent to the following variant of (13)

$$\delta_{\mathrm{ub}}(x, Z) = \min_{\mathcal{A}} \{\delta : \nabla f(x) = 0, \quad \nabla^2 f(x) \succeq 0, \quad \mathcal{A} \text{ satisfies } (n, \delta)\text{-RIP}\}, \tag{14}$$

with optimal value $\delta_{\mathrm{ub}}(x, Z) = (1 - \eta^\star)/(1 + \eta^\star)$. Now, (14) gives an upper-bound on (13) because $(n, \delta)$-RIP is a *sufficient* condition for $(2r, \delta)$-RIP. Hence, we have $\delta_{\mathrm{ub}}(x, Z) \geq \delta^\star$ for every valid choice of $x$ and $Z$.

The same convex formulation can be modified to bound $\delta^\star$ from below[2]. Specifically, a necessary condition for $\mathcal{A}$ to satisfy $(2r, \delta_{2r})$-RIP is the following

$$(1 - \delta_{2r})\|UYU^T\|_F^2 \leq \|\mathcal{A}(UYU^T)\|^2 \leq (1 + \delta_{2r})\|UYU^T\|_F^2 \qquad \forall Y \in \mathbb{R}^{2r \times 2r} \tag{15}$$

where $U$ is a *fixed* $n \times 2r$ matrix. This is a convex linear matrix inequality; substituting (15) into (13) in lieu of of $(2r, \delta)$-RIP yields a convex optimization problem

$$\delta_{\mathrm{lb}}(x, Z, \mathcal{U}) = \min_{\mathcal{A}} \{\delta : \nabla f(x) = 0, \quad \nabla^2 f(x) \succeq 0, \quad (15)\}, \tag{16}$$

that generates lower-bounds $\delta^\star \geq \delta_{\mathrm{lb}}(x, Z, U)$.

**Our best upper-bound is likely $\delta^\star \leq 1/2$.** The existence of Example 3 gives the upper-bound of $\delta^\star \leq 1/2$. To improve upon this bound, we randomly sample $x, z \in \mathbb{R}^{n \times r}$ i.i.d. from the standard Gaussian, and evaluate $\delta_{\mathrm{ub}}(x, zz^T)$ using MOSEK [2]. We perform the experiment for 3 hours on each tuple $(n, r) \in \{1, 2, \ldots, 10\} \times \{1, 2\}$ but obtain $\delta_{\mathrm{ub}}(x, zz^T) \geq 1/2$ for every $x$ and $z$ considered.

**The threshold is likely $\delta^\star = 1/2$.** Now, we randomly sample $x, z \in \mathbb{R}^{n \times r}$ i.i.d. from the standard Gaussian. For each fixed $\{x, z\}$, we set $U = [x, z]$ and evaluate $\delta_{\mathrm{lb}}(x, Z, U)$ using MOSEK [2]. We perform the same experiment as the above, but find that $\delta_{\mathrm{lb}}(x, zz^T, U) \geq 1/2$ for every $x$ and $z$ considered. Combined with the existence of the upper-bound $\delta^\star = 1/2$, these experiments strongly suggest that $\delta^\star = 1/2$.

## 5 Experiment 2: SGD escapes spurious local minima

How is the performance of SGD affected by the presence of spurious local minima? Given that spurious local minima cease to exist with $\delta < 1/5$, we might conjecture that the performance of SGD is a decreasing function of $\delta$. Indeed, this conjecture is generally supported by evidence from

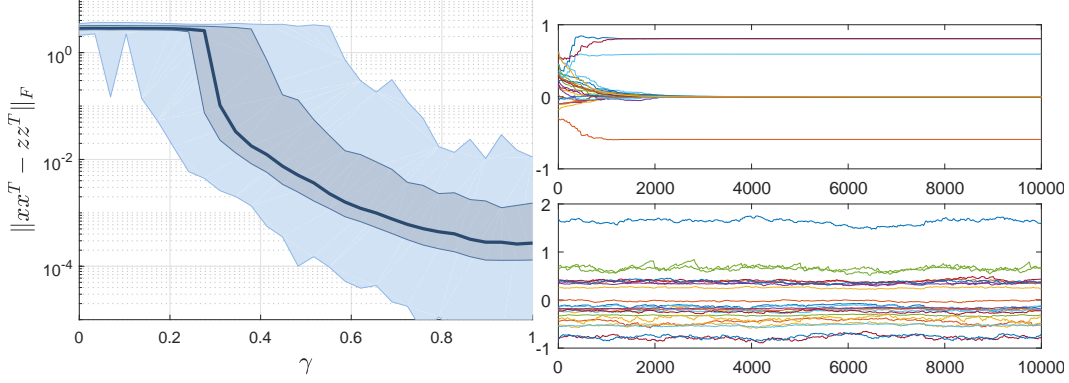

Figure 2: "Bad" instance ($n = 12$, $r = 2$) with RIP constant $\delta = 0.973$ and spurious local min at $x_{loc}$ satisfying $\|xx^T\|_F / \|zz^T\|_F \approx 4$. Here, $\gamma$ controls initial SGD $x = \gamma w + (1 - \gamma)x_{loc}$ where $w$ is random Gaussian. **(Left)** Error distribution after 10,000 SGD steps (rate $10^{-4}$, momentum 0.9) over 1,000 trials. Line: median. Inner bands: 5%-95% quantile. Outer bands: min/max. **(Right top)** Random initialization with $\gamma = 1$; **(Right bottom)** Initialization at local min with $\gamma = 0$.

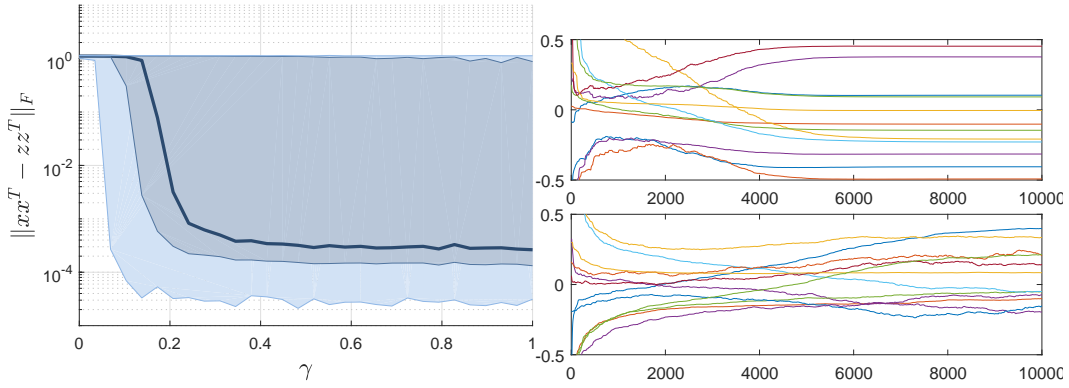

Figure 3: "Good" instance ($n = 12$, $r = 1$) with RIP constant $\delta = 1/2$ and spurious local min at $x_{loc}$ satisfying $\|xx^T\|_F / \|zz^T\|_F = 1/2$ and $x^T z = 0$. Here, $\gamma$ controls initial SGD $x = \gamma w + (1 - \gamma)x_{loc}$ where $w$ is random Gaussian. **(Left)** Error distribution after 10,000 SGD steps (rate $10^{-3}$, momentum 0.9) over 1,000 trials. Line: median. Inner bands: 5%-95% quantile. Outer bands: min/max. **(Right top)** Random initialization $\gamma = 1$ with success; **(Right bottom)** Random initialization $\gamma = 1$ with failure.

the nearly-isotropic measurement ensembles [6, 5, 32, 1], all of which show improving performance with increasing number of measurements $m$.

This section empirically measures SGD (with momentum, fixed learning rates, and batchsizes of one) on two instances of (1) with different values of $\delta$, both engineered to contain spurious local minima by numerically solving (10). We consider a "bad" instance, with $\delta = 0.975$ and rank $r = 2$, and a "good" instance, with $\delta = 1/2$ and rank $r = 1$. The condition number of the "bad" instance is 25 times higher than the "good" instance, so classical theory suggests the former to be a factor of 5-25 times harder to solve than the former. Moreover, the "good" instance is locally strongly convex at its isolated global minima while the "bad" instance is only locally weakly convex, so first-order methods like SGD should locally converge at a linear rate for the former, and sublinearly for the latter.

**SGD consistently succeeds on "bad" instance with $\delta = 0.975$ and $r = 2$.** We generate the "bad" instance by fixing $n = 12$, $r = 2$, selecting $x, z \in \mathbb{R}^{n \times r}$ i.i.d. from the standard Gaussian, rescale $z$ so that $\|zz^T\|_F = 1$ and rescale $x$ so that $\|xx^T\|_F / \|zz^T\|_F \approx 4$, and solving (10); the results are shown in Figure 2. The results at $\gamma \approx 0$ validate $x_{loc}$ as a true local minimum: if initialized here, then SGD remains stuck here with $> 100\%$ error. The results at $\gamma \approx 1$ shows randomly initialized

SGD either escaping our engineered spurious local minimum, or avoiding it altogether. All 1,000 trials at $\gamma = 1$ recover the ground truth to $< 1\%$ accuracy, with 95% quantile at $\approx 0.6\%$.

**SGD consistently fails on "good" instance with $\delta = 1/2$ and $r = 1$.** We generate the "good" instance with $n = 12$ and $r = 1$ using the procedure in the previous Section; the results are shown in Figure 3. As expected, the results at $\gamma \approx 0$ validate $x_{loc}$ as a true local minimum. However, even with $\gamma = 1$ yielding a random initialization, 59 of the 1,000 trials still result in an error of $> 50\%$, thereby yielding a failure rate of $5.90 \pm 2.24\%$ up to three standard deviations. Examine the failed trials closer, we do indeed find SGD hovering around our engineered spurious local minimum.

Repeating the experiment over other instances of (1) obtained by solving (10) with randomly selected $x, z$, we generally obtain graphs that look like Figure 2. In other words, SGD usually escapes spurious local minima even when they are engineered to exist. These observations continue to hold true with even massive condition numbers on the order of $10^4$, with corresponding RIP constant $\delta = 1 - 10^{-4}$. On the other hand, we do occasionally sample well-conditioned instances that behave closer to the "good" instance describe above, causing SGD to consistently fail.

# 6   Conclusions

The nonconvex formulation of low-rank matrix recovery is highly effective, despite the apparent risk of getting stuck at a spurious local minimum. Recent results have shown that if the linear measurements of the low-rank matrix satisfy a restricted isometry property (RIP), then the problem contains *no spurious local minima*, so exact recovery is guaranteed. Most of these existing results are based on a norm-preserving argument: relating $\|\mathcal{A}(xx^T - Z)\| \approx \|xx^T - Z\|_F$ and arguing that a lack of spurious local minima in the latter implies a similar statement in the former.

Our key message in this paper is that moderate RIP is not enough to eliminate spurious local minima. To prove this, we formulate a convex optimization problem in Section 2 that generates counterexamples that satisfy RIP but contain spurious local minima. Solving this convex formulation in closed-form in Section 3 shows that counterexamples are ubiquitous: almost any rank-1 $Z \succeq 0$ and any $x \in \mathbb{R}^n$ can respectively be the ground truth and spurious local minimum to an instance of matrix recovery satisfying RIP. We gave one specific counterexample with RIP constant $\delta = 1/2$ in the introduction that causes randomly initialized stochastic gradient descent (SGD) to fail 12% of the time.

Moreover, stochastic gradient descent (SGD) is often but not always able to avoid and escape spurious local minima. In Section 5, randomly initialized SGD solved one example with a 100% success rate over 1,000 trials, despite the presence of spurious local minima. However, it failed with a consistent rate of $\approx 6\%$ on another other example with an RIP constant of just $1/2$. Hence, as long as spurious local minima exist, we cannot expect to guarantee exact recovery with SGD (without a much deeper understanding of the algorithm).

Overall, exact recovery guarantees will generally require a proof of no spurious local minima. However, arguments based solely on norm preservation are conservative, because most measurements are not isotropic enough to eliminate spurious local minima.

## Acknowledgements

We thank our three NIPS reviewers for helpful comments and suggestions. In particular, we thank reviewer #2 for a key insight that allowed us to lower-bound $\delta^\star$ in Section 4. This work was supported by the ONR Awards N00014-17-1-2933 and ONR N00014-18-1-2526, NSF Award 1808859, DARPA Award D16AP00002, and AFOSR Award FA9550- 17-1-0163.

## Footnotes

[1]Given a linear map, the condition number measures the ratio in size between the largest and smallest images, given a unit-sized input. Within our specific context, the $2r$-restricted condition number is the smallest $\kappa_{2r} = L/\ell$ such that $\ell\|X\|_F^2 \leq \|\mathcal{A}(X)\|^2 \leq L\|X\|_F^2$ holds for all rank-$2r$ matrices $X$.

[2]We thank an anonymous reviewer for this key insight.

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
