[Supplementary Material]

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

$$\mathscr{L}^T : \mathbb{R}^{n \times r} \to \mathbb{S}^{n^2} \qquad \mathscr{L}^T(y) = e y^T \mathbf{X}^T + \mathbf{X} y e^T$$

and also

$$\mathscr{M} : \mathbb{S}^{n^2} \to \mathbb{S}^{nr} \qquad \mathscr{M}(\mathbf{H}) = 2 \cdot [I_r \otimes \mathrm{mat}(\mathbf{H}e)] + \mathbf{X}^T \mathbf{H} \mathbf{X},$$
$$\mathscr{M}^T : \mathbb{S}^{nr} \to \mathbb{S}^{n^2} \qquad \mathscr{M}^T(U) = \mathrm{vec}\,(U)e^T + e\mathrm{vec}\,(U)^T + \mathbf{X} U \mathbf{X}^T.$$

Moreover, we use $\rho = \|x\| / \|z\|$ and $\phi = \arccos(x^T z / \|x\| \|z\|)$.

### A.1    Technical lemmas

We begin by solving an eigenvalue LMI in closed-form.

**Lemma 9.** *Given $M \in \mathbb{S}^n$ with $\mathrm{tr}(M) \geq 0$, we split the matrix into a positive part $M_+$ and a negative part $M_-$ satisfying*

$$M = M_+ - M_- \qquad where \qquad M_+, M_- \succeq 0, \quad M_+ M_- = 0.$$

*Then the following problem has solution*

$$\mathrm{tr}(M_-)/\mathrm{tr}(M_+) = \min_{\substack{\alpha \in \mathbb{R} \\ U,V \succeq 0}} \{\mathrm{tr}(V) : \mathrm{tr}(U) = 1, \alpha M = U - V\}$$

*Proof.* Write $p^\star$ as the optimal value. Then,

$$
\begin{aligned}
p^\star &= \max_{\beta} \min_{\substack{\alpha \in \mathbb{R} \\ U,V \succeq 0}} \{\mathrm{tr}(V) + \beta \cdot [\mathrm{tr}(U) - 1] : \alpha M = U - V\} \\
&= \max_{\beta \geq 0} \min_{\alpha \in \mathbb{R}} \{-\beta + \min_{U,V \succeq 0} \{\mathrm{tr}(V) + \beta \cdot \mathrm{tr}(U) : \alpha M = U - V\}\} \\
&= \max_{\beta \geq 0} \min_{\alpha \in \mathbb{R}} \{-\beta + \alpha \cdot [\mathrm{tr}(M_-) + \beta \cdot \mathrm{tr}(M_+)]\} \\
&= \max_{\beta \geq 0} \{-\beta : \mathrm{tr}(M_-) + \beta \cdot \mathrm{tr}(M_+) = 0\} \\
&= \mathrm{tr}(M_-)/\mathrm{tr}(M_+).
\end{aligned}
$$

The first line converts an equality constraint into a Lagrangian. The second line isolates the optimization over $U, V \succeq 0$ with $\beta \geq 0$, noting that $\beta < 0$ would yield $\mathrm{tr}(U) \to \infty$. The third line solves the minimization over $U, V \succeq 0$ in closed-form. The fourth line views $\alpha$ as a Lagrange multiplier. □

The matrix $\mathscr{L}^T(y)$ is rank-2 with the following eigenvalues.

**Lemma 10.** *The matrix $\mathscr{L}^T(y)$ is rank-2, and its two nonzero eigenvalues are*

$$\|\mathbf{X}y\|\|e\|(\cos\theta_y \pm 1), \quad \text{where } \cos\theta_y = \frac{e^T\mathbf{X}y}{\|e\|\|\mathbf{X}y\|}. \tag{17}$$

*Proof.* We project $\mathbf{X}y$ onto $e$ and define $q$ as the residual, as in $\mathbf{X}y = \alpha e + q$ with $\alpha = (e^T\mathbf{X}y)/\|e\|^2$. Then we have the similarity relation

$$\mathscr{L}^T(y) = \begin{bmatrix} e & q \end{bmatrix} \begin{bmatrix} 2\alpha & 1 \\ 1 & 0 \end{bmatrix} \begin{bmatrix} e & q \end{bmatrix}^T \sim \|e\| \cdot \begin{bmatrix} 2\alpha\|e\| & \|q\| \\ \|q\| & 0 \end{bmatrix},$$

and the $2\times2$ matrix has eigenvalues $\|\alpha e\|^2 \pm \sqrt{\|\alpha e\|^2 + \|q\|^2}$. Substituting $\|\mathbf{X}y\|^2 = \|\alpha e\|^2 + \|q\|^2$ completes the proof. $\quad\square$

Also, the angle between $e$ and $\text{range}(\mathbf{X})$ is closely associated with the angle between $x$ and $z$.

**Lemma 11.** *Define the incidence angle $\theta$ between $e$ and $\text{range}(\mathbf{X})$ as*

$$\theta = \arccos\left(\max_y \frac{e^T\mathbf{X}y}{\|e\|\|\mathbf{X}y\|}\right). \tag{18}$$

*Then, the angle has value*

$$\sin\theta = \frac{(\|z\|\sin\phi)^2}{\|e\|} = \frac{\sin^2\phi}{\sqrt{(\rho^2-1)^2 + 2\rho^2\sin^2\phi}}.$$

*Proof.* We project $z$ onto $\text{range}(x)$ and define $w$ as the residual, as in $z = x\alpha + w$ where $\alpha = (x^Tz)/\|x\|^2$. Then, we have the similarity relation

$$xx^T - zz^T = \begin{bmatrix} x & w \end{bmatrix} \begin{bmatrix} (1-\alpha^2)I_r & -\alpha I_r \\ -\alpha I_r & -I_r \end{bmatrix} \begin{bmatrix} x & w \end{bmatrix}^T \sim \begin{bmatrix} (1-\alpha^2)\|x\|^2 & -\alpha\|x\|\|w\| \\ -\alpha\|x\|\|w\| & -\|w\|^2 \end{bmatrix},$$

and may solve the problem of projecting $e$ onto $\text{range}(\mathbf{X})$ after a change of basis

$$\begin{aligned}
\|e\|\sin\theta &= \min_y \|\mathbf{X}y - e\| \\
&= \min_y \|xy^T + yx^T - (xx^T - zz^T)\|_F, \\
&= \min_{\tilde{y}_1,\tilde{y}_2} \left\| \begin{bmatrix} \tilde{y}_1 & \tilde{y}_2 \\ \tilde{y}_2 & 0 \end{bmatrix} - \begin{bmatrix} (1-\alpha^2)\|x\|^2 & -\alpha\|x\|\|w\| \\ -\alpha\|x\|\|w\| & -\|w\|^2 \end{bmatrix} \right\|_F, \\
&= \|w\|^2 = \|z\|^2\sin^2\phi.
\end{aligned}$$

This proves the first equality. On the other hand, we have

$$\|e\| = \|xx^T - zz\|_F = \sqrt{\|x\|^4 + \|z\|^4 - 2(x^Tz)^2} = \|z\|^2\sqrt{\rho^4 + 1 - 2\rho^2\cos\phi}. \tag{19}$$

Completing the square and substituting yields the second equality. $\quad\square$

**Lemma 12.** *Let $\hat{\mathbf{H}}$ be the optimal choice in Theorem 6. Then*

$$\|\text{mat}(\hat{\mathbf{H}}e)\| \le \sqrt{1+\rho^4}\|z\|^2, \qquad \lambda_{\min}(\mathbf{X}^T P_{e\perp}\mathbf{X}) \ge 2\|x\|^2\zeta^2$$

*where $\zeta$ was defined in (12).*

*Proof.* For the first bound, we have

$$u^T\text{mat}(\hat{\mathbf{H}}e)u = (u\otimes u)^T\hat{\mathbf{H}}e \le \|u\otimes u\|\|\hat{H}\|\|e\| = \|u\|^2\|e\|,$$

and $\|e\|^2 = \|z\|^4(1 - \rho^2\cos\phi + \rho^4) \leq \|z\|^4(1 + \rho^4)$ from (19). For the second bound, define $\theta$ as the angle between $e$ and $\mathrm{range}(\mathbf{X})$ in (11), and note that $\zeta$ in (12) satisfies $\zeta = \sin\theta$ by construction via Lemma 11. Then,

$$v^T(\mathbf{X}^T P_{e\perp}\mathbf{X})v = \|P_{e\perp}\mathbf{X}v\|^2 \text{ because projections are idempotent: } P_{e\perp} = P_{e\perp}^2$$

$$= \min_{\alpha\in\mathbb{R}}\|\mathbf{X}v - e\alpha\|^2 = \min_{\alpha\in\mathbb{R}}\{\|\mathbf{X}v\|^2 - 2\alpha e^T\mathbf{X}v + \alpha^2\|e\|^2\}$$

$$\geq \min_{\alpha\in\mathbb{R}}\{\|\mathbf{X}v\|^2 - 2\alpha\|e\|\|\mathbf{X}v\|\cos\theta + \alpha^2\|e\|^2\}$$

$$\text{whose minimum is attained at } \alpha = \|\mathbf{X}v\|\cos\theta$$

$$= \|\mathbf{X}v\|^2(1 - \cos^2\theta) = \|\mathbf{X}v\|^2\sin^2\theta,$$

and

$$\|\mathbf{X}v\|^2 = \|xv^T + vx^T\|_F^2 = 2\|x\|^2\|v\|^2 + 2(x^Tv)^2 \geq 2\|x\|^2\|v\|^2.$$

Finally, dividing by $\|v\|^2$ yields the desired bound. $\qquad\square$

## A.2   Proof of Theorem 6

The problem of finding the best-conditioned $\mathbf{H}$ satisfying $\mathscr{L}(\mathbf{H}) = 0$ is the following primal-dual LMI pair

$$\begin{array}{ll} \underset{\mathbf{H},\eta}{\text{maximize }} \eta & \qquad\qquad \underset{y,U_1,U_2}{\text{minimize }} \mathrm{tr}(U_2) \qquad\qquad\qquad (20) \\[1mm] \text{subject to } \mathscr{L}(\mathbf{H}) = 0, & \qquad\qquad \text{subject to } \mathscr{L}^T(y) = U_1 - U_2, \\[1mm] \qquad\qquad \eta I \preceq \mathbf{H} \preceq I. & \qquad\qquad\qquad \mathrm{tr}(U_1) = 1, \quad U_1, U_2 \succeq 0, \end{array}$$

where $\mathscr{L}^T$ is the adjoint operator to $\mathscr{L}$ in (7). Slater's condition is trivially satisfied by the dual: $y = 0$ and $U_1 = U_2 = \nu^{-1}I$ with $\nu = \frac{1}{2}n(n+1)$ is a strictly feasible point. Hence, strong duality holds, meaning that the two objectives coincide with $\mathrm{tr}(U_2^\star) = \eta^\star$ at optimality, so we implicitly solve the primal by solving the dual.

The mechanics of the dual problem become more obvious if we first optimize over $U_1$ and $U_2$ and the length of $y$. Applying Lemma 9 yields

$$\underset{y}{\text{minimize }} \frac{\sum_{i=1}^n (-\lambda_i(\mathscr{L}^T(y)))_+}{\sum_{i=1}^n (+\lambda_i(\mathscr{L}^T(y)))_+} \qquad \text{where} \qquad (\alpha)_+ \equiv \begin{cases} \alpha & \alpha \geq 0 \\ 0 & \alpha < 0 \end{cases}. \qquad (21)$$

The goal of this latter problem is to find a vector $y$ that maximizes the sum of the positive eigenvalues of $\mathscr{L}^T(y)$, while minimizing the (absolute) sum of the negative eigenvalues. In Lemma 10, we prove that $\mathscr{L}^T(y)$ has exactly one positive eigenvalue and one negative eigenvalue, and their values in the rank-1 case are closely related to the angle $\phi$ between $x$ and $z$. Substituting this into (21) yields an unconstrained minimization

$$\underset{y}{\text{minimize }} \frac{1 - \cos\theta_y}{1 + \cos\theta_y} \qquad \text{where} \qquad \cos\theta_y = \frac{e^T\mathbf{X}y}{\|e\|\|\mathbf{X}y\|}.$$

In turn, Lemma 11 yields $\max_y \cos\theta_y = \cos\theta = \sqrt{1 - \sin^2\theta}$ where $\sin\theta \equiv \zeta$ in the statement of Theorem 6.

## A.3   Proof of Theorem 7

We show that $\mathbf{H}_\tau \equiv \tau P_{e\perp} + \mathbf{H}_0$ with some $\tau \geq 0$ is a feasible point for (10) with a small condition number. Here, $P_{e\perp} = I - ee^T/\|e\|^2$ is the projection onto the kernel of $e$, and $\mathbf{H}_0 \preceq I$ is the best-conditioned $\mathbf{H} \succeq 0$ satisfying $\mathscr{L}(\mathbf{H}) = 0$ from Theorem 6. Observe that $\mathrm{cond}(\mathbf{H}_\tau) = (1 + \tau)\cdot\mathrm{cond}(\mathbf{H}_0)$.

Let us find the smallest $\tau \geq 0$ to guarantee that $\mathscr{L}(\mathbf{H}_\tau) = 0$ and $\mathscr{M}(\mathbf{H}_\tau) \succeq \mu I$, for some choice of $\mu > 0$. Note that $\mathscr{L}(\mathbf{H}_\tau) = 0$ is satisfied by construction, because $\mathscr{L}(\mathbf{H}_\tau) = \tau\mathscr{L}(P_{e\perp}) + (1 - \tau)\mathscr{L}(\mathbf{H}_0)$, and $\mathscr{L}(\mathbf{H}_0) = 0$ by hypothesis while $\mathscr{L}(P_{e\perp}) = 2\mathbf{X}^T(P_{e\perp})e = 0$. Hence, our only difficulty is finding the smallest $0 \leq \tau < 1$ such that

$$\mathscr{M}(\mathbf{H}_\tau) = 2\mathrm{mat}(\mathbf{H}_0 e) + \tau\mathbf{X}^T P_{e\perp}\mathbf{X}^T + \mathbf{X}^T\mathbf{H}_0\mathbf{X}^T \succ 0.$$

In Lemma 12, we prove the following two inequalities

$$\|\mathrm{mat}(\mathbf{H}_0 e)\| \leq \sqrt{1+\rho^4}\|z\|^2, \qquad \lambda_{\min}(\mathbf{X}^T P_{e\perp}\mathbf{X}) \geq 2\|x\|^2\zeta^2$$

Hence, $\mathscr{M}(\mathbf{H}_\tau) \succeq \mu I$ with $\mu = \sqrt{1+\rho^4}\|z\|^2 \geq \|z\|^2$ is guaranteed if we set

$$\frac{4\|\mathrm{mat}(\mathbf{H}_0 e)\|}{\lambda_{\min}(\mathbf{X}^T P_{e\perp}\mathbf{X})} \leq \frac{4\sqrt{1+\rho^4}\|z\|^2}{2\|x\|^2\zeta^2} = \frac{2\sqrt{\rho^2 + \rho^{-2}}}{\zeta^2} = \tau.$$

Rescaling $\mathbf{H}_\tau$ by $1/(1+\tau)$ completes the proof for the feasibility statement.

Finally, to derive the RIP constant bound $\delta_{2r} \leq (\tau + \sqrt{1-\zeta^2})/(\tau+1)$, write $\delta \equiv \sqrt{1-\zeta^2}$ and note that we have

$$\eta_2^\star \geq \frac{1-\delta}{1+\tau} \cdot \frac{1}{1+\delta} = \left(1 - \frac{\tau+\delta}{1+\tau}\right) \cdot \frac{1}{1+\delta} \geq \frac{1-(\tau+\delta)/(\tau+1)}{1+(\tau+\delta)/(\tau+1)},$$

where the last bound is due to the fact that $(\tau+\delta)/(\tau+1) \geq \delta$ holds for all $\tau, \delta \geq 0$. Multiplying through by $1 + (\tau+\delta)/(\tau+1)$ yields

$$\left(1 - \frac{\tau+\delta}{\tau+1}\right)\|X\|_F^2 \leq \left(1 + \frac{\tau+\delta}{\tau+1}\right)\|\mathcal{A}(X)\|_F^2 \leq \left(1 + \frac{\tau+\delta}{\tau+1}\right)\|X\|_F^2$$

for all $X$.