[Reviews · NeurIPS 2018]

Reviewer 1



Post rebuttal comment: I have read the authors feedback. I recommend the authors to add the discussion about the sharpness of the results to the manuscript in a new section. This is a crucial part which improves the manuscript clarity. ======= There are many existing results in the literature that many recovery problems behave "nice" under restricted isometry property (RIP). This paper has an interesting view of looking at the problem in the other direction, i.e., how much RIP is needed so that all local optima be globally optimal. The paper shows that the RIP is necessary by providing examples that the nice property does not hold for moderate RIP levels. In the reviewer's opinion, the paper tackles an important questions. However, the following concerns prevents the reviewer from giving higher ratings to the paper: -- Many relevant works, e.g. G. Tang et al., is missed and the paper should be placed with respect to these existing results. -- It is not clear whether the RIP levels provided in this work are the optimal sharp bounds.

Reviewer 2



Summary It has recently been proved that simple non-convex algorithms can recover a low-rank matrix from linear measurements, provided that the measurement operator satisfies a Restricted Isometry Property, with a good enough constant. Indeed, in this case, the natural objective function associated with the problem has no second-order critical point other than the solution. In this article, the authors ask how good the constant must be so that this property holds. They propose a convex program that finds measurement operators satisfying RIP, but for which the objective function has "bad" second-order critical points. They analyze this program in the case where the matrix to be recovered has rank 1, and deduce from this analysis that, for any x,z, there exists a RIP operator such that x is a bad second-order critical point when the true unknown is zz*; they upper bound the RIP constant. Finally, they run their program, and study through a few examples the behavior of Stochastic Gradient Descent on instances generated by it. Main remarks The main things I appreciated in this article are: 1. I found the topic really interesting. There has been a lot of works, in the recent years, on why simple non-convex heuristics are able to efficiently solve some a priori difficult problems. Understanding which properties the problems must satisfy so that this is possible is an almost completely open problem. The present article studies, and relatively precisely describes, an aspect of this question. 2. I appreciated the approach used to tackle the problem, with this convex program that allows for both a theoretical and a numerical analysis. It was new to me, and proved efficient to answer the quesiton solved by the authors. 3. The writing style is clear and easy to read. However, there are also, in my opinion, some issues: 1. I think the proof of Theorem 7 should absolutely be corrected and rewritten. I do not say that it is not true, but there are many things in it that I do not understand: - I do not see why naively repeating the prof of Theorem 5 leads to the equation after line 422, because I do not see why the positive eigenvalues of L^T(y)-M^T(U3) should be essentially the same as in Lemma 8 (the presence of X U3 X^T bothers me). Additionally, u3 should be defined. - Lines 431-432: it is g(0)=sin theta and g'(0)=cos theta, so I do not get the equation that follows. - In Lemma 10, it seems to me that g(beta) should be defined as a max, not a min (minimizing the objective function is equivalent to maximizing cos theta). - In the equation after line 457, why is it "+ V2 beta" and not "- V2 beta"? And I do not understand the second equality. Do we have alpha^2 + beta^2 = 1? - Line 458: Why is it sufficient to show the orthogonality? - Line after 459: A line of explanation as to why the optimal y and U solve this problem would be welcome, and the \tilde u_i should be defined. 2. Some aspects of the presentation could be improved. In particular, - Any measurement operator that is injective on the set of rank-r matrices satisfies 2r-RIP for some constant stricly smaller than one. But given that it is in general NP-hard to recover a low-rank matrix from linear measurements, it is somewhat expected that, for some RIP matrices, spurious second-order critical points (or at least points that look very much like spurious second-order critical points) exist. Consequently, it seems to me that it is not a surprise that the answer to the question on lines 58-59 is negative. Subsection 1.1 does not make this clear. - It is a naive remark, but 0 is always a critical point of the objective function; the discussion after Theorem 5 seems to ignore this. - Subsection 1.3: There are proofs of correctness for non-convex methods that do not really rely on properties very close to RIP. This is notably the case in phase retrieval, where the RIP does not hold. 3. It seems to me that some natural questions raised by this work could have been explored. - Experiment 2 shows that SGD avoids local minima more or less often, depending on their configuration. It would have been interesting to suggest explanations as to the factors that control this phenomenon. If I understood correctly, an obvious difference between the first and second examples is that, in the first case, the spurious local minimum has norm 2, while in the second one, it has norm 1/sqrt(2). The initialization strategy gives more weight to vectors with small norm. Does this explain part of the difference between the two examples? Can the autors propose and test other hypotheses? - About the tightness of the bound in Experiment 1, it seems to me that it should be possible to get a lower bound on the optimal delta, by replacing, in Equation (10), the equation "eta I <= H <= I" by "eta I <= P^T H P <= I", where the columns of P form an orthonormal basis of a subspace of S^n whose matrices all have rank at most 2r (for example, the matrices in S^n whose range is included in the vector space generated by the ranges of x and z). Is it correct? Have the authors tested it? Minor remarks - Theorem 2: "local minimum" -> "second-order critical point" - Do the authors have ideas on how to generate operators A that satisfy the same properties, but for which m < n(n+1)/2? - Subsection 1.3: It is surprising to say that convex approaches enjoy stronger guarantees, then that some can be shown to always succeed when delta_(4r) < 0.4142 or delta_(4r) < 0.1907 (conditions which are not really weaker that requiring delta_(2r) < 0.2). - Line 132: "require" -> "requires" - The formulation in line 156 slightly suggests that the condition number is a convex function of H (which is not the case). - There are typos in Equation (6) (an "A" and a factor 2 missing, I think). - Equation (10): isn't the condition H \succeq 0 redundant? - In Section 3 and in the associated proofs, it would be better to remove all occurences of the variable "r", since only the r=1 case is studied. - Line 222: It seems to me that Example 3 only shows that the smallest delta achievable is smaller than or equal to 1/2, not that it is exactly 1/2. - Right sides of Figures 2 and 3: what does the y axis represent? - Line 243: "the the" -> "the". - Bibliography: "Candès" sometimes has an accent, sometimes not. - Line 387: There is a word missing after "coincide". - Lines 389-392: I do not think it is true that U1 takes on the positive eigenvalues, and U2 the negative ones. Imagine that Lambda = Diag(0.5,-0.5). Then U1 = V Diag(1,0) V^T and U2 = V Diag(0.5,0.5) V^T satisfy the constraints, and minimize the trace of U2. - Line 395: "eigenavlues" -> "eigenvalues" - Equation after Line 398: there is a theta missing in the denominator. - Line 436: the expression for M^T is not correct. - Line 442: there are typos in the expression for the eigenvalues. - Theorem 5 and Lemma 9: What is the interest of introducing the lower bound sin^2 phi / sqrt(1+rho^4) for sin theta? The exact expression that comes out of Lemma 9 is not so much more complex. - Equation after Line 450: There is a transpose missing. - Equation after Line 464: There are typos. - Equations in the proof of Lemma 11: v should be defined. Added after author feedback: Many thanks to the authors for their clear and detailed response. Almost all of my remarks seem to have been taken into account. I changed my score accordingly.

Reviewer 3



This paper studies restricted isometry property (RIP) that is a commonly required condition in matrix recovery. It shows that a small RIP constant is required for the loss function not to have spurious local minima. It proposes an optimization method to find a matrix recovery problem that has a reasonably small RIP constant and has spurious local minima. This paper also empirically justified the common observation that SGD can escape spurious local minima in most cases. The results in this paper shows the different behaviors of SGD in different cases, which shows the complication of SGD analyses. In general, the results of this paper are interesting and, I believe, will provide further insights into areas like matrix recovery, SGD analyses, and non-convex optimization analyses. However, the main contribution of this paper is only the observation that RIP constant should be small to rule out spurious local minima, which a little bit weak. I'd expect more studies in this direction. For example, it would be interesting to narrow the gap between the current maximal theoretical RIP constant 1/5 and the minimal empirical RIP constant 1/2. Also, it would be great to develop other substitutions of RIP that allow a larger range of feasible problems. A minor question: In Line 64-65, it is claimed that delta = 1/5 is too restrictive. Is there a quantitative way to describe the restrictiveness? === post-rebuttal review === I've read the rebuttal and it addresses my question. I also increased the score to 7.